# Exocentric-to-Egocentric Video Generation

**Jia-Wei Liu**[1*], **Weijia Mao**[1*], **Zhongcong Xu**[1], **Jussi Keppo**[2], **Mike Zheng Shou**[1✉]

[1]Show Lab, [2]National University of Singapore

Figure 1: Given sparse $4$ exocentric videos configured 360° around daily-life skilled human activities such as playing basketball (upper), CPR training (lower), our Exo2Ego-V can generate corresponding egocentric videos with the same activity and environment as the exocentric videos. We encourage readers to click and play the video clips in this figure using Adobe Acrobat.

## Abstract

We introduce Exo2Ego-V, a novel exocentric-to-egocentric diffusion-based video generation method for daily-life skilled human activities where sparse 4-view exocentric viewpoints are configured 360° around the scene. This task is particularly challenging due to the significant variations between exocentric and egocentric viewpoints and high complexity of dynamic motions and real-world daily-life environments. To address these challenges, we first propose a new diffusion-based multi-view exocentric encoder to extract the dense multi-scale features from multi-view exocentric videos as the appearance conditions for egocentric video generation. Then, we design an exocentric-to-egocentric view translation prior to provide spatially aligned egocentric features as a concatenation guidance for the input of egocentric video diffusion model. Finally, we introduce the temporal attention layers into our egocentric video diffusion pipeline to improve the temporal consistency cross egocentric frames. Extensive experiments demonstrate that Exo2Ego-V significantly outperforms SOTA approaches on $5$ categories from the Ego-Exo4D dataset with an average of $35\%$ in terms of LPIPS. Our code and model will be made available on `https://github.com/showlab/Exo2Ego-V`.

## 1 Introduction

When people observe and learn skills such as cooking or playing basketball from an exocentric (third-person) perspective, they can easily envision themselves executing these skills from an egocentric (first-person) perspective [2]. This exocentric-egocentric (Exo-Ego) translation remains the foundation of visual learning [15] for both human beings and AI robots [8, 4, 3, 37, 35], and unleashes new opportunities for AI assistant [54, 7, 13] and augmented reality [1]. However, it remains particularly challenging for computer vision algorithms to achieve such exocentric to egocentric video generation for daily-life skilled human activities, primarily due to 1) significant variations between exocentric

---

[*] Equal Contribution    [✉] Corresponding Author

38th Conference on Neural Information Processing Systems (NeurIPS 2024).

and egocentric viewpoints, and 2) high complexity of dynamic motions and daily-life environments, as illustrated in Fig. 1.

Existing view translation approaches mainly focus on the task of novel view synthesis and have made remarkable progress particularly since the introduction of Neural Radiance Fields (NeRF) [36]. While initially limited to reconstructing static 3D scenes, subsequent studies have extended NeRF to address the challenges of dynamic view synthesis [42, 38, 39, 50, 27, 12, 30]. However, these approaches are limited to per-scene regressive optimization and require dozens or hundreds of views as input. As a result, they fall short in the exocentric to egocentric video generation task due to the sparse yet highly variable viewpoints and significant occlusions.

The remarkable success of powerful image diffusion models [45] provides new opportunities for introducing such generative models in the task of exo-ego generation. Recent attempt [33] leverages action intention consisting of human movement and action description for Ego2Exo video generation. However, it strictly requires the first exocentric frame a priori which largely simplifies the ego-to-exo generation task to optical flow prediction and exocentric frame warping, highly limiting its applications in general exo-ego generation tasks. On the other hand, Exo2Ego [34] attempts to tackle with the exo2ego view translation by first transferring exo hand pose to ego, and then learning the conditional distribution of the target ego image given a single exo image and the predicted ego hand pose. Despite promising, it is limited to image-level translation and requires a carefully designed capturing setup with the exocentric camera configured close to the hand-object region and thus is restricted to desktop activities with simple environments.

In contrast, we contend that it is necessary for exo-ego translation algorithms to be resilient to the complexity and diversity of daily-life scenarios such as cooking in kitchens, playing basketball in courts, etc. The introduction of Ego-Exo4D [15] opens new opportunities and challenges for exo-ego translation by providing a large-scale simultaneously-captured egocentric and exocentric videos of daily-life skilled human activities. In order to capture the complete and complex human-environment activities, they configure 4 exocentric cameras in 360° around the dynamic scene, resulting in new challenges of significant variations between exocentric and egocentric viewpoints, as well as complex dynamic motions and daily-life environments, as shown in Fig. 1.

To tackle with these challenges, we propose a novel exocentric-to-egocentric video diffusion pipeline dubbed as Exo2Ego-V. We address the significant challenges of large viewpoint variations and complex environments from two aspects: exo appearance conditions and ego translation prior. Firstly, we propose a diffusion-based multi-view exocentric encoder to extract the multi-scale exocentric features as the appearance conditions for egocentric video generation. We achieve this by concatenating ego hidden states with exo features for self-attention computation, so that the ego hidden states can attend to both the egocentric features as well as the multi-view exocentric features through the self-attention mechanism. In addition, we inject the relative position information into our exocentric encoder by adding exocentric latents with relative Exo2Ego relative camera pose embedding. Our exocentric encoder can extract dense human activity and environment information to guide the appearance of egocentric video generation pipeline. Secondly, we introduce an Exo2Ego view translation prior based on PixelNeRF [60] to provide coarse yet spatially aligned egocentric features as a concatenation guidance for the input of egocentric video diffusion model. Finally, to improve the temporal dynamic motion consistency of egocentric video contents, we insert temporal layers into our egocentric video diffusion pipeline to encode the temporal information across ego frames.

We extensively evaluate our Exo2Ego-V on 5 categories of skilled human activities from the challenging Ego-Exo4D [15] dataset and H2O dataset [26]. As shown in Fig. 1, our Exo2Ego-V can generate the corresponding egocentric videos given 4 multi-view exocentric videos, and significantly outperforms SOTA approaches with an average of 35% in terms of LPIPS.

To summarize, the major contributions of our paper are:

- We present a novel framework of Exo2Ego-V, the first work to achieve exocentric-to-egocentric video generation for daily-life real-world skilled human activities.

- We propose a new diffusion-based multi-view exocentric encoder and an Exo2Ego view translation prior that can extract dense exocentric features and spatially aligned egocentric features as conditions for our egocentric video diffusion pipeline.

- Extensive experiments show that Exo2Ego-V significantly outperforms SOTA approaches on the challenging Ego-Exo4D [15] dataset with an average of 35% in terms of LPIPS.

## 2 Related work

### 2.1 Egocentric-exocentric vision

Tremendous progress has been made for exocentric vision with various visual perception and generation tasks due to the large amount of dataset captured at thrid-person views [11, 23, 48]. Recently, egocentric vision has also been scaling up particularly since the introduction of EPIC-Kitchens [9, 10] and Ego4D [14]. Previous attempts for joint egocentric and exocentric vision explore learning the ego-exo view-invariant features on paired small-scale dataset [2, 61] or through unpaired learning [59]. Another line of research focuses on egocentric human localization from exocentric videos [57, 52], as well as egocentric human pose estimation from exocentric videos [51]. More recently, the introduction of Ego-Exo4D dataset [15] opens up new opportunities for joint egocentric and exocentric vision with large-scale synchronized multi-view ego-exo videos with multi-modality annotations.

### 2.2 Egocentric-exocentric cross-view translation

**Ego-exo view translation.** There is limited prior work on ego-exo cross-view translation. Early attempt [28] explores the exo-to-ego image generation with a novel parallel generative adversarial network to learn shared features of exo and ego images. STA-GAN [29] further extends P-GAN [28] to Exo2Ego video synthesis with a spatial temporal attention fusion module. However, they are limited to simple activities such as walking where most contents in egocentric and exocentric views are static environments [29, 28]. More recently, IDE [33] leverages action intention consisting of human movement and action description for ego2exo video generation. However, it requires the first exocentric frame a priori which largely simplifies the ego-to-exo generation task to optical flow prediction and exocentric frame warping. On the other hand, Exo2Ego [34] attempts to tackle with the exo2ego view translation by first transferring exo hand pose to ego, and then learning the conditional distribution of the target ego image given a single exo image and predicted ego hand pose. Despite promising, it is limited to image-level translation and requires a carefully designed capturing setup with the exocentric camera configured close to the hand-object region and thus is restricted to desktop activities with simple environments.

**Novel view synthesis (NVS).** Ego-Exo view translation is also related to NVS, which has made remarkable progress particularly since the introduction of NeRF [36]. While initially limited to reconstructing static 3D scenes, NeRF has been extended to modelling dynamic scenes [42, 38, 39, 50, 27, 12], dynamic humans [41, 53, 22, 32]. However, these approaches are limited to per-scene regressive optimization and require dozens or hundreds of views as input. On the other hand, generalizable scene reconstruction methods [60, 56] are still limited to static scenes. As a result, they fall short in the exocentric to egocentric video generation task due to the sparse yet highly variable viewpoints and significant occlusions.

### 2.3 Video generation

Recent works have extended the power of image diffusion models to video editing [55, 43, 31] and generation [6, 62, 63, 18, 20]. Tune-A-Video [55] inflates the image diffusion with cross-frame attention and fine-tunes the source video, aiming to implicitly learn the source motion and transfer it to the target video. Video Diffusion Models(VDM) [20] designs a factorized space-time UNet to generate videos. Stable Video Diffusion [5] introduces a systematic data curation workflow, enabling the training of a state-of-the-art text-to-video and image-to-video models. AnimateDiff [16] proposes a plug-and-play motion module on temporal layers for personalized text-to-image animation. Other approaches [21, 58] introduce such video generation architectures to human image animation and achieve faithful performances. Our Exo2Ego-V is a new video diffusion pipeline for the challenging Exo2Ego generation on daily-life skilled human activities.

## 3 Method

### 3.1 Preliminaries

**Latent diffusion models (LDMs).** LDMs encode input images to a latent representation using a pretrained variational auto-encoder (VAE) and operate the diffusion and denoising process following

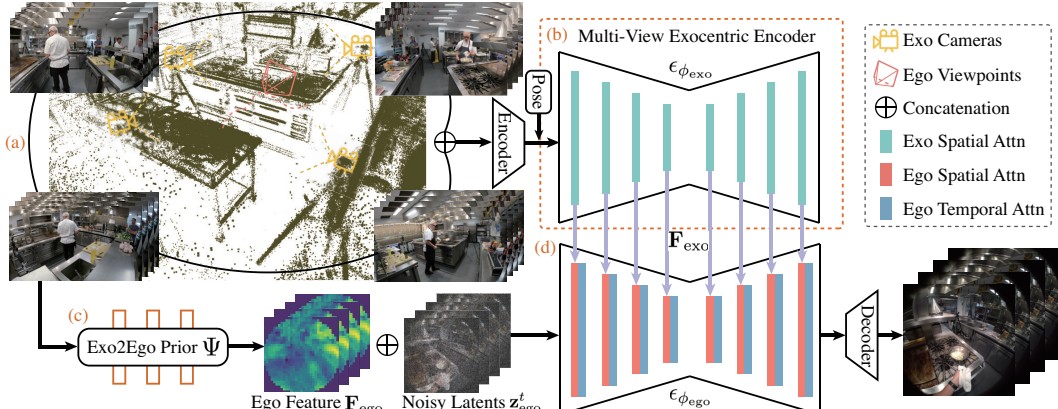

Figure 2: **Overview of Exo2Ego-V.** Given 4 exocentric videos configured 360° around daily-life skilled human activities such as cooking (a), our multi-view exocentric encoder (b) extracts the multi-scale exocentric features as the appearance conditions for egocentric video generation, and our Exo2Ego view translation prior (c) predicts the egocentric features as the concatenation guidance for the egocentric noisy latents input. With these information, our egocentric video diffusion pipeline (d) generates the egocentric videos with the same activity and environment as the exocentric videos.

denoising diffusion probabilistic models (DDPMs) [19] (see Sec. A.6 for more details) in the reduced-dimension latent space, and finally decode the denoised latents to the image space. Our model is an extention of the pretrained text-to-image latent diffusion model [45] that utilizes the UNet architecture [49] for denoising noise prediction with multiple down, middle, and up blocks. Each block consists of a ResNet2D layer, a self-attention layer, and a cross-attention layer.

## 3.2 Overall framework

**Task definition.** Given 4 exocentric videos $\mathbf{V}_{\text{exo}} = \left[ \mathbf{V}_{\text{exo}}^1, \mathbf{V}_{\text{exo}}^2, \mathbf{V}_{\text{exo}}^3, \mathbf{V}_{\text{exo}}^4 \right]$ configured 360° around daily-life skilled human activities, our objective is to generate their corresponding egocentric video $\mathbf{V}_{\text{ego}}$, as shown in Fig. 2(a). Since these daily-life activities happen naturally in real-world scenarios such as kitchens, basketball courts, bike stores, etc, our task features the diversity of the real world with complex activities and environments. Therefore, the sparse 4 exocentric cameras have to be configured evenly in 360° around the dynamic scene in order to capture both the complex human activities and real-world environments [15], resulting in **significant variations between exocentric and egocentric viewpoints**. Furthermore, such real-world skilled human activities such as cooking, repairing bike, and playing basketball are also highly challenging in terms of the **complexity of dynamic motions and real-world environments**.

**Overall framework.** In order to tackle with the above challenges, we propose a novel exocentric-to-egocentric video diffusion pipeline dubbed as Exo2Ego-V, as shown in Fig. 2. To address the significant challenges of large viewpoint variations and complex environments, we propose a diffusion-based multi-view exocentric encoder (Fig. 2(b)) to extract the multi-scale multi-view exocentric features as the appearance conditions for egocentric video generation. In addition, we design an exocentric-to-egocentric view translation prior (Fig. 2(c)) based on PixelNeRF [60] to provide coarse yet spatially aligned egocentric features as a concatenation guidance for egocentric video generation. Finally, we introduce the temporal attention layers into our egocentric video diffusion pipeline to improve the temporal consistency cross egocentric frames (Fig. 2(d)).

## 3.3 Multi-view exocentric encoder

**Motivation.** Our task is featured with the significant variations of Exo2Ego viewpoints and complex environments. The core to tackle with this challenge is to fully explore the multi-view exocentric information for the purpose of guiding the egocentric video generation. A naive solution is to utilize the CLIP [44] image encoder to extract latent features from low-resolution images. However, such semantic-level CLIP image features fall short in extracting the dense and fine-grained detail information from exocentric videos. Another naive solution is to train a 4D dynamic scene reconstruc-

tion model for each multi-view exocentric sequences, but it requires high computation and storage resources and current methods cannot handle sparse 4 views complex dynamic scene reconstruction.

**Framework.** Inspired by recent reference image animation methods [58, 21] that extract dense image features with a reference UNet to preserve reference human identity, we propose our multi-view exocentric encoder with a different purpose of extracting the dense multi-view exocentric intricate details as appearance conditions for egocentric video generation. Specifically, our exocentric encoder creates a trainable copy of the base Ego UNet and inject the relative camera poses from exocentric viewpoints to the egocentric viewpoints as additional embeddings, as shown in Fig. 2(b). We additionally explore adding temporal layers for exocentric encoder in our ablation study.

Given $\mathbf{V}_{\text{exo}} \in \mathbb{R}^{N \times C \times F \times H \times W}$ and their relative camera poses $\mathbf{P} \in \mathbb{R}^{N \times 4 \times 4}$, our exocentric encoder computes the multi-view multi-scale appearance condition features $\mathbf{F}_{\text{exo}}$ for the egocentric video generation at denoising step $t = 0$:

$$\mathbf{F}_{\text{exo}} = \epsilon_{\phi_{\text{exo}}} \left( \mathbf{z}_{\text{exo}}^t; \mathbf{V}_{\text{exo}}, \mathbf{P}, t \right) , \tag{1}$$

where $N = 4$ is the number of exocentric views and $F = 8$ is the number of frames for each video. $\mathbf{F}_{\text{exo}}$ are the normalized attention features for the downsampling, middle, and upsampling blocks of Exo UNet. We set $t = 0$ to preserve the appearance details of the noise-free exocentric videos.

Then, the exocentric features $\mathbf{F}_{\text{exo}}$ are utilized as the appearance conditions for egocentric video generation by concatenating $\mathbf{F}_{\text{exo}}$ with the corresponding egocentric UNet hidden states $\mathbf{z}_{\text{ego}}^t$ for the self-attention layers in every block $b$ at each denoising step $t$:

$$\mathbf{Q}_b = \mathbf{W}_b^{\mathbf{Q}} \cdot \mathbf{z}_{\text{ego},b}^t, \ \mathbf{K}_b = \mathbf{W}_b^{\mathbf{K}} \cdot \left[ \mathbf{z}_{\text{ego},b}^t, \mathbf{F}_{\text{exo},b} \right], \ \mathbf{V}_b = \mathbf{W}_b^{\mathbf{V}} \cdot \left[ \mathbf{z}_{\text{ego},b}^t, \mathbf{F}_{\text{exo},b} \right] , \tag{2}$$

where $[\cdot]$ denotes concatenation operation, and the Ego UNet self-attention is: $\text{Softmax} \left( \frac{\mathbf{Q}_b \cdot \mathbf{K}_b^{\mathbf{T}}}{\sqrt{d}} \right) \cdot \mathbf{V}_b$. The queried egocentric noisy latents can attend to both the egocentric features as well as the multi-view exocentric features through the self-attention mechanism, and thus translate the appearance of complex human skill activities and environments from exocentric views to the egocentric view.

**Camera pose.** In order to inject the relative position information into our Exo2Ego generation pipeline, inspired by MVDream [47], we embed the relative Exo2Ego camera poses with a 2-layer MLP and add the embedding with the denoising timestep embedding for our multi-view exocentric encoder. Since we set $t = 0$ for our Exo UNet, the relative camera pose embeddings are the main embedding to inject the relative position information into the exocentric feature extraction.

### 3.4 Exocentric-to-egocentric view translation prior

**Motivation.** Although our multi-view exocentric encoder can extract dense and intricate appearance details, it entirely relies on the self-attention mechanism to explore the correspondences from the egocentric contents to the exocentric features, which are still challenging for our scenarios with large viewpoints variations. To tackle with this, we design an Exo2Ego view translation prior based on PixelNeRF [60] to generate a coarse yet spatially aligned egocentric latent feature as the concatenation guidance for our egocentric video generation pipeline.

**Framework.** Given multi-view exocentric videos $\mathbf{V}_{\text{exo}}$, exo camera poses $\mathbf{P}_{\text{exo}}$, egocentric videos $\mathbf{V}_{\text{ego}}$, and ego camera poses $\mathbf{P}_{\text{ego}}$, we learn an Exo2Ego view translation prior $\Psi$ by training a generalizable PixelNeRF [60] for all timesteps. For each synchronized timestep of the 4 exocentric videos and 1 egocentric video, at each iteration we extract 4 exo frames and 1 ego frame and randomly sample rays from these 5 images for optimization. Inspired by ReconFusion [56], we utilize a light PixelNeRF with 6-layer MLPs for higher efficiency. Please see Sec. A.1 for more details.

As shown in Fig. 2(c), with this Exo2Ego translation prior, we can render both the egocentric features $\mathbf{F}_{\text{ego}}$ and egocentric pixels $\mathbf{I}_{\text{ego}}$ given the multi-view exocentric videos, exo camera poses, and queried ego camera pose:

$$\Psi \left( \mathbf{V}_{\text{exo}}, \mathbf{P}_{\text{exo}}, \mathbf{P}_{\text{ego}} \right) \longmapsto \left( \mathbf{F}_{\text{ego}}, \mathbf{I}_{\text{ego}} \right), \tag{3}$$

Inspired by ReconFusion [56], we design our translation prior to render egocentric features $\mathbf{F}_{\text{ego}}$ at the egocentric viewpoint with the same spatial resolution as the egocentric latents, so that $\mathbf{F}_{\text{ego}}$ is spatially aligned with the noisy egocentric latents. Therefore, we concatenate $\mathbf{z}_{\text{ego}}^t$ with $\mathbf{F}_{\text{ego}}$ along channel dimension as the input to the egocentric video diffusion model to predict the noise $\epsilon_{\text{ego}}$ at

each denoising timestep $t$. In addition, we extract the CLIP image feature of the rendered ego images $\mathbf{I}_{\text{ego}}$ as the cross-attention information for our egocentric video diffusion pipeline:

$$\epsilon_{\text{ego}} = \epsilon_{\phi_{\text{ego}}} \left( \left[ \mathbf{z}_{\text{ego}}^t, \mathbf{F}_{\text{ego}} \right] ; \mathbf{F}_{\text{exo}}, \mathbf{I}_{\text{ego}}, t \right) . \tag{4}$$

### 3.5 Temporal dynamic motion layer

To improve the temporal dynamic motion consistency of egocentric and exocentric video contents, we follow common practice [18, 20] to insert temporal attention layers within the 2D UNet blocks, as shown in Fig. 2(d). Specifically, we insert the temporal layers on the egocentric video generation pipeline and we additionally ablate on inserting the temporal layers on the exocentric encoder. As such, the input egocentric latents $\mathbf{z}_{\text{ego}}^t \in \mathbb{R}^{N \times C \times F \times H \times W}$ are first reshaped to $\mathbb{R}^{(NF) \times H \times W \times C}$ for computing the spatial attentions with egocentric and exocentric features in spatial layers, and then reshaped to $\mathbb{R}^{(NHW) \times F \times C}$ to compute the temporal cross-frame information in temporal layers.

### 3.6 Optimization

We optimize our Exo2Ego-V in a 2-stage training strategy. In the first stage, we remove the temporal layers and optimize the Exo2Ego spatial appearance translation modules, including the multi-view exocentric encoder, Exo2Ego view translation prior, as well as the Ego UNet. In the second stage, we only optimize the Ego temporal layers for temporal consistency and freeze other modules.

**Exo2Ego spatial appearance translation.** We first pre-train our Exo2Ego view translation prior with the pixel-level reconstruction loss $\mathcal{L}_{\text{REC}}$. Then, we alternately finetune the Exo2Ego view translation prior with the reconstruction loss $\mathcal{L}_{\text{REC}}$, and multi-view exocentric encoder and the Ego UNet with the noise prediction loss $\mathcal{L}_{\text{S}}$.

$$\mathcal{L}_{\text{REC}} = \| \mathbf{R}_{\text{render}} - \mathbf{R}_{\text{gt}} \|_2^2 , \quad \mathcal{L}_{\text{S}} = \mathbb{E}_{\mathbf{z}_{\text{ego}}^t, \mathbf{V}_{\text{exo}}, \mathbf{P}_{\text{exo}}, t, \epsilon} \left[ \omega\left(t\right) \| \epsilon - \epsilon_{\text{ego}} \|_2^2 \right] , \tag{5}$$

where $\mathbf{R}_{\text{render}}$ is the rendered ray pixels sampled randomly from exocentric and egocentric frames, $\mathbf{R}_{\text{gt}}$ is the corresponding ground-truth pixels. $\epsilon$ is the ego noise sampled from $\mathcal{N}(0, 1)$. $w(t)$ is a weighting function that depends on the noise level $t$.

**Temporal motion finetuning.** In the second stage, we freeze the translation prior and Exo and Ego UNets, and finetune the pretrained temporal layers from AnimateDiff [16] on our egocentric and exocentric videos with $F$ frames in temporal dimension.

$$\mathcal{L}_{\text{T}} = \mathbb{E}_{\mathbf{z}_{\text{ego}}^{t,F}, \mathbf{V}_{\text{exo}}^F, \mathbf{P}_{\text{exo}}^F, t, \epsilon^F,} \left[ \omega^F\left(t\right) \| \epsilon^F - \epsilon_{\text{ego}}^F \|_2^2 \right] . \tag{6}$$

## 4 Experiments

### 4.1 Dataset

We evaluate our method on 5 categories of Ego-Exo4D dataset [15] featuring both exocentric and egocentric human activities: Cooking, Covid Test, Basketball, CPR, and Bike. Each category contains synchronized captured exo and ego videos of different participants performing these activities at different locations around the world. Specifically, **Cooking** captures people preparing various dishes in kitchens. **Basketball** captures participants playing basketball in courts. **Covid Test** captures individuals conducting covid tests for themselves in various scenes. **CPR** captures scenes where participants perform cardiopulmonary resuscitation on a CPR model. **Bike** captures scenes of participants repairing bikes in bike stores. We set the the number of temporal frames to 8 and spatial resolution to $480 \times 270$ and $256 \times 256$ for exocentric and egocentric videos, respectively. For each video from the above categories, we extract frames at 7.5 fps and split them into multiple action clips according to the action annotations and turning timesteps where the participant's head pose turns more than $45°$ within 1 second. We retain the videos that contain both ego and exo intrinsic and extrinsic parameters. Finally, we processed 489 videos from Cooking category, 909 videos from Basketball category, 127 videos from Covid Test category, 66 videos from CPR category, and 359 videos from Bike category. The videos lengths vary between $3 \sim 15$ minutes for different categories. We also evaluate our method on the H2O dataset [26], which provides synchronized multi-view Exo-Ego images for desktop activities.

We train our Exo2Ego-V (see Sec. A.2 for training details) and baselines for each category and utilize the following 3 test evaluation: (1) **Unseen action**: We split each video into multiple clips based on the action annotations, so that each clip features a different action. We use $80\%$ of action clips as our train set and the remaining $20\%$ unseen action clips as test set. (2) **Unseen take**: Each take refers to a complete human activity video. Each participant conduct $2 \sim 4$ takes for an activity. We randomly select one take as our test set and use the remaining takes for training. (3) **Unseen scenes**: Each category is captured in multiple different scenes around the world. We randomly select one entire scene with multiple takes out of a category as our test set and use the remaining scenes for training.

## 4.2 Comparisons with SOTA approaches

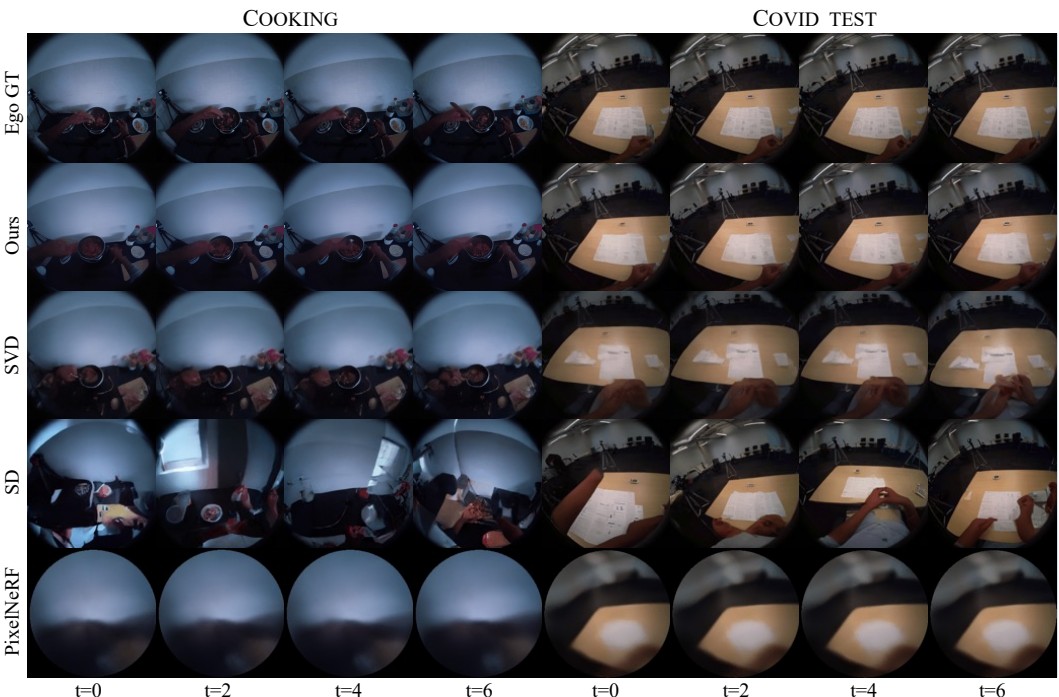

Figure 3: Qualitative comparisons of our method against SOTA approaches on **unseen actions**.

**Baselines.** We compare our Exo2Ego-V with three baselines. (1) *Stable Video Diffusion (SVD)* [5], a recent state-of-the-art image-to-video diffusion model. (2) *Stable Diffusion (SD)* [46], a powerful text-to-image diffusion model. (3) *PixelNeRF* [60], a general 3D scene reconstruction model. We adapt SVD and SD for our Exo2Ego generation task by inputting 4 exo views as their conditions and train the models to generate ego views. Specifically, we first use a VAE model to obtain the latent contents of each exo view and concatenate ego noisy latent with these 4 exo latents along the channel dimension as input. Then, we use the CLIP model to obtain the exo image CLIP features as the cross-attention information for SVD and SD. We train these three baselines for each categories.

**Quantitative results.** We report PSNR, SSIM, and LPIPS with AlexNet [25] that measure the differences between generated ego frames and groundtruth on Tab. 1. Our Exo2Ego-V achieves the best performance for both unseen actions and unseen takes in terms of all metrics. It is noted PixelNeRF [60] achieves good PSNR scores since PSNR favors blurry images [38] as shown in Fig. 3 and 4. Most importantly, our Exo2Ego-V significantly outperforms SOTA approaches on all categories with an average of $35\%$ in terms of LPIPS, which clearly demonstrates the superiority of our Exo2Ego-V. We additionally evaluate our Exo2Ego-V and best-performing baseline SVD [5] on the H2O

Table 2: Quantitative comparison of our method against SVD on H2O dataset.

|  | PSNR↑ | SSIM↑ | LPIPS ↓ |
|---|---|---|---|
| SVD [5] | 16.530 | 0.468 | 0.271 |
| Ours | 18.600 | 0.581 | 0.189 |

dataset [26] in Tab. 2. Our model still achieves the best performance, which demonstrate the generalizability of our method on different datasets.

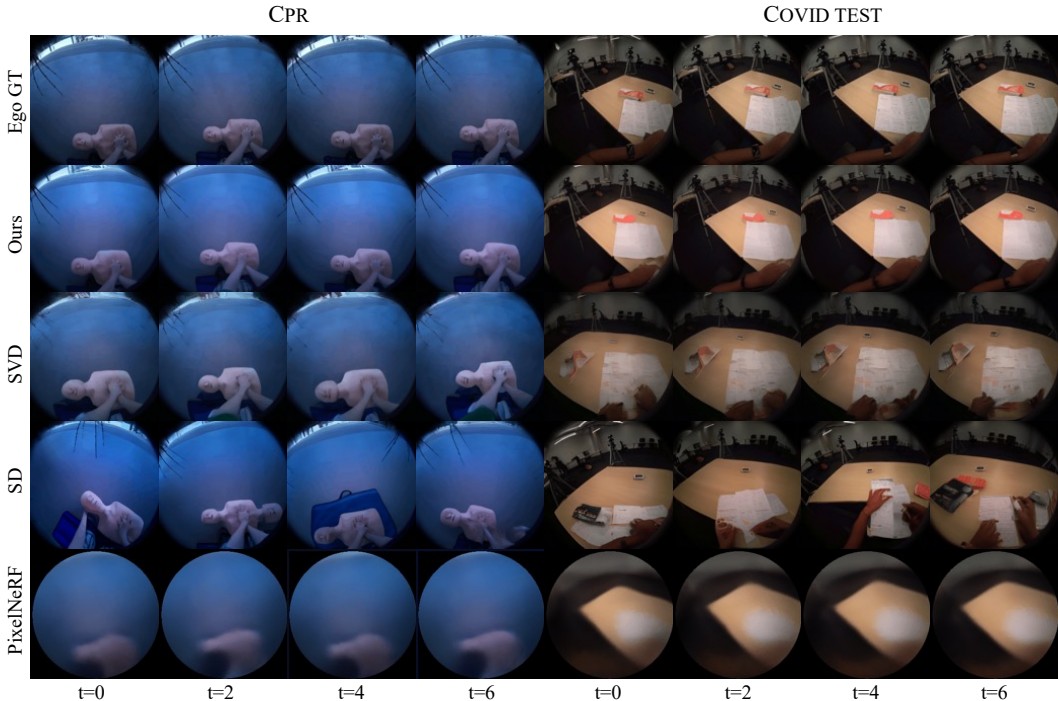

Figure 4: Qualitative comparisons of our method against SOTA approaches on **unseen takes**.

Table 1: Averaged quantitative evaluation on different categories. We color code each cell as best .

| | UNSEEN ACTION | | | | | | | | | | | | | | |
| | COOKING | | | BASKETBALL | | | COVID TEST | | | CPR | | | BIKE | | |
| | PSNR↑ | SSIM↑ | LPIPS↓ | PSNR↑ | SSIM↑ | LPIPS↓ | PSNR↑ | SSIM↑ | LPIPS↓ | PSNR↑ | SSIM↑ | LPIPS↓ | PSNR↑ | SSIM↑ | LPIPS↓ |
|---|---|---|---|---|---|---|---|---|---|---|---|---|---|---|---|
| PixelNeRF [60] | 17.278 | 0.412 | 0.640 | 18.054 | 0.512 | 0.599 | 19.707 | 0.548 | 0.543 | 18.444 | 0.561 | 0.558 | 16.070 | 0.351 | 0.679 |
| SD [46] | 12.167 | 0.313 | 0.583 | 12.480 | 0.400 | 0.605 | 14.200 | 0.413 | 0.538 | 16.543 | 0.573 | 0.454 | 12.510 | 0.289 | 0.577 |
| SVD [5] | 14.318 | 0.407 | 0.519 | 15.529 | 0.491 | 0.533 | 16.584 | 0.507 | 0.477 | 17.807 | 0.630 | 0.397 | 14.541 | 0.364 | 0.516 |
| Ours | 17.367 | 0.493 | 0.408 | 20.062 | 0.624 | 0.249 | 21.462 | 0.668 | 0.235 | 18.533 | 0.647 | 0.305 | 16.310 | 0.413 | 0.486 |
| | UNSEEN TAKE | | | | | | | | | | | | | | |
| | COOKING | | | BASKETBALL | | | COVID TEST | | | CPR | | | BIKE | | |
| | PSNR↑ | SSIM↑ | LPIPS↓ | PSNR↑ | SSIM↑ | LPIPS↓ | PSNR↑ | SSIM↑ | LPIPS↓ | PSNR↑ | SSIM↑ | LPIPS↓ | PSNR↑ | SSIM↑ | LPIPS↓ |
| PixelNeRF [60] | 17.177 | 0.426 | 0.644 | 18.744 | 0.549 | 0.600 | 20.402 | 0.561 | 0.531 | 16.674 | 0.634 | 0.476 | 16.129 | 0.393 | 0.663 |
| SD [46] | 12.542 | 0.324 | 0.582 | 13.048 | 0.433 | 0.612 | 14.654 | 0.416 | 0.541 | 12.510 | 0.667 | 0.356 | 12.510 | 0.314 | 0.581 |
| SVD [5] | 14.554 | 0.416 | 0.532 | 16.563 | 0.546 | 0.555 | 16.875 | 0.500 | 0.490 | 18.302 | 0.680 | 0.353 | 14.492 | 0.383 | 0.527 |
| Ours | 17.712 | 0.504 | 0.456 | 21.417 | 0.646 | 0.300 | 21.590 | 0.667 | 0.245 | 18.473 | 0.711 | 0.219 | 16.343 | 0.441 | 0.489 |

**Qualitative results.** Fig. 3 and 4 visualizes the qualitative comparison of Exo2Ego-V over SOTA approaches on unseen actions and unseen takes, respectively, where Exo2Ego-V achieves substantially better egocentric videos quality than other approaches for all categories (see Sec. A.3 for more results). SVD [5] and SD [46] encounter significant difficulties by conditioning on the highly semantic exo images features to generate egocentric videos. In addition, SD [46] falls short in temporal consistency due to its 2D image-level generation. PixelNeRF [60] renders very blurry results due to the significant difficulty of sparse yet highly variable viewpoints and large occlusions. In addition, Fig. 6 visualizes the comparison of our method against SVD [5] on the H2O dataset [26]. Our method achieves the best performance and generates photorealistic hand-object interactions. Please see **supplementary video** for more results on video comparisons, which demonstrates the superiority of our Exo2Ego-V on both much higher spatial appearance quality and temporal consistency compared to other methods.

Table 3: Averaged quantitative evaluation on different categories against baselines for **unseen scenes**.

| | UNSEEN SCENE | | | | | | | | | | | |
| | COOKING | | | BASKETBALL | | | COVID TEST | | | CPR | | |
| | PSNR↑ | SSIM↑ | LPIPS↓ | PSNR↑ | SSIM↑ | LPIPS↓ | PSNR↑ | SSIM↑ | LPIPS↓ | PSNR↑ | SSIM↑ | LPIPS↓ |
|---|---|---|---|---|---|---|---|---|---|---|---|---|
| PixelNeRF [60] | 13.393 | 0.355 | 0.700 | 16.091 | 0.418 | 0.676 | 14.885 | 0.438 | 0.697 | 14.349 | 0.348 | 0.745 |
| SD [46] | 10.617 | 0.268 | 0.587 | 11.686 | 0.334 | 0.640 | 13.072 | 0.409 | 0.624 | 14.937 | 0.428 | 0.586 |
| SVD [5] | 11.960 | 0.321 | 0.553 | 14.468 | 0.385 | 0.586 | 14.392 | 0.484 | 0.627 | 14.934 | 0.447 | 0.580 |
| Ours | 13.926 | 0.389 | 0.602 | 16.201 | 0.462 | 0.560 | 14.127 | 0.387 | 0.604 | 15.387 | 0.553 | 0.654 |

**Comparisons on unseen scenes.** We conduct additional experiments on unseen scenes of our Exo2Ego-V and baselines, and report PSNR, SSIM, and LPIPS with AlexNet [25] that measure the differences between generated ego frames and groundtruth on Tab. 3. We do not conduct experiments on Bike category since it is only captured on 4 different scenes, which are too few to generalize to new scenes. Fig. 5 visualizes the qualitative comparison of our Exo2Ego-V over SOTA approaches on unseen scenes. Our Exo2Ego-V achieves the best performance on most metrics in Tab. 3 and substantially better egocentric videos quality than other approaches as shown in Fig. 5. PixelNeRF [60] still renders very blurry results but gets good PSNR values since PSNR favors blurry images [38]. We also find that it is very challenging for all methods to evaluate on the unseen scenes due to the significant variance of new environments compared to the training set, and the lack of large-scale scene diversity from the training data.

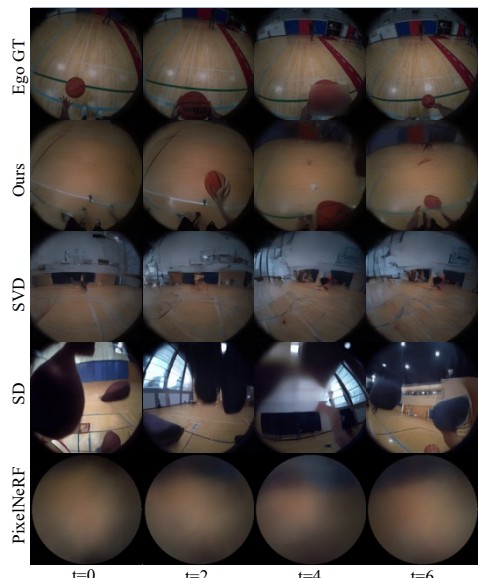

Figure 5: Qualitative comparisons of our method against SOTA approaches on **unseen scenes**.

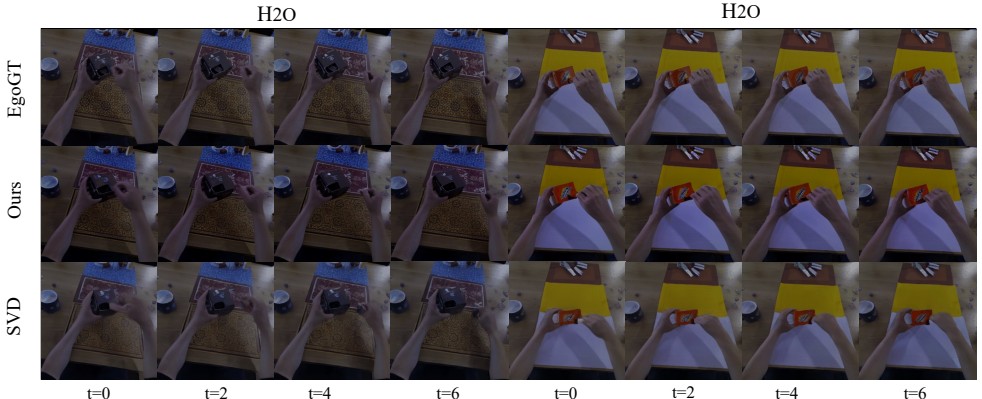

Figure 6: Qualitative comparisons of our method against SOTA approaches on H2O dataset.

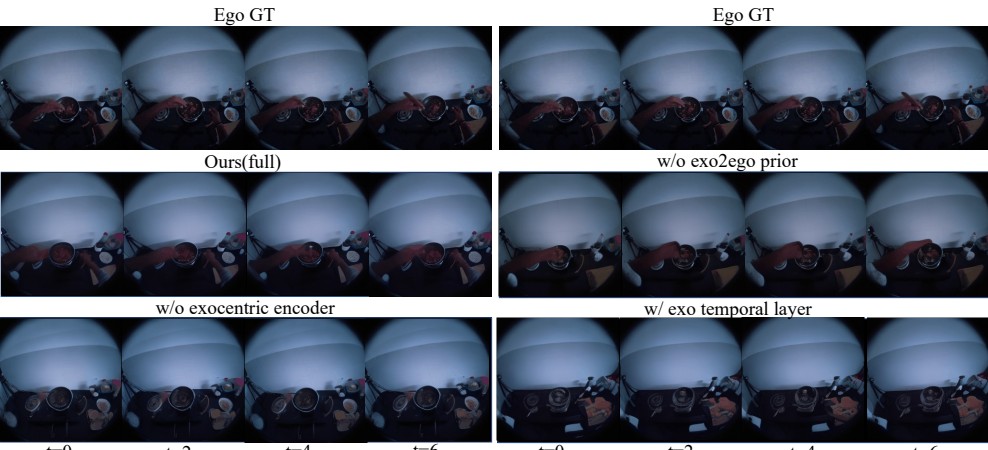

Figure 7: Qualitative ablation results of our method for cooking category on unseen actions.

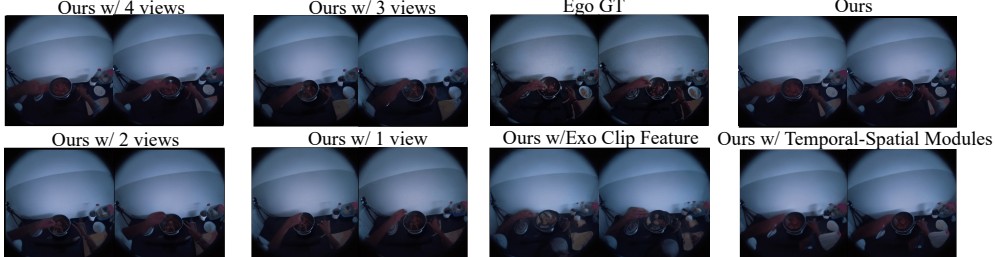

Figure 8: More ablation results of our method for cooking category on unseen actions.

## 4.3 Ablation study

We conduct ablation studies on the cooking category from Ego-Exo4D dataset [15]. We ablate on the proposed multi-view exocentric encoder, Exo2Ego view translation prior, and the temporal layers of multi-view exocentric encoder. As shown in Tab. 4, our full model achieves the best performance in terms of PSNR and SSIM. In addition, we provide the qualitative results of our ablations in Fig. 7 , which further demonstrates the effectiveness of our designs. Removing exocentric encoder results in inferior performance than full model, which clearly proves its capability in extracting dense and multi-scale exo features for ego video generation. Although removing exo2ego prior achieves the best LPIPS, it results in a cleaner but inaccurate egocentric video due to the lack of egocentric guidance, which improves the LPIPS but gets a lower PSNR. As shown in Fig. 7, removing exo2ego prior results in the missing of right arm. In addition, we ablate on adding temporal layers for our multi-view exocentric encoder and evaluate the temporal consistency by computing the CLIP image embeddings on our generated ego clips and report the average cosine similarity between all pairs of clip frames. Adding the exo temporal layers achieves a higher averaged temporal score of $0.924$ compared to $0.918$ of full model for unseen actions, demonstrating higher temporal consistency but at the expense of inferior image-level quality in Tab. 4. Thus, we disgard the exo temporal layer in final model.

We ablate on the number of exo views and replacing our exocentric feature encoder with CLIP features in Tab. 4 and Fig. 8. Our model with 4 exo views achieves the best performance, and our method achieves much better performance compared to the one using CLIP features. We also ablate on first performing temporal attention and then spatial attention for our model. The spatial-temporal model is slightly better than the temporal-spatial model in terms of PSNR and SSIM, and slightly worse for LPIPS. We follow the spatial-temporal attentions [16, 5].

Table 4: Ablation results of our method.

| | COOKING | | | | | |
| | UNSEEN ACTION | | | UNSEEN TAKE | | |
| | PSNR↑ | SSIM↑ | LPIPS↓ | PSNR↑ | SSIM↑ | LPIPS↓ |
|---|---|---|---|---|---|---|
| w/ Exo temporal layer | 16.762 | 0.473 | 0.507 | 16.784 | 0.489 | 0.584 |
| w/o Exocentric encoder | 16.777 | 0.455 | 0.441 | 16.734 | 0.450 | 0.499 |
| w/o Exo2ego prior | 17.268 | 0.493 | 0.364 | 17.668 | 0.502 | 0.401 |
| w/ 3 Views | 17.230 | 0.486 | 0.383 | 17.400 | 0.489 | 0.428 |
| w/ 2 Views | 16.930 | 0.474 | 0.399 | 17.020 | 0.479 | 0.445 |
| w/ 1 Views | 17.020 | 0.478 | 0.395 | 17.200 | 0.476 | 0.439 |
| w/ Exo CLIP | 16.540 | 0.456 | 0.425 | 16.410 | 0.445 | 0.480 |
| w/ Temporal-spatial | 17.000 | 0.484 | 0.402 | 17.290 | 0.490 | 0.443 |
| Ours (full) | 17.367 | 0.493 | 0.408 | 17.712 | 0.504 | 0.456 |

## 5 Conclusion

We introduced a novel framework of Exo2Ego-V, the first work to achieve Exo2Ego video generation for daily-life real-world skilled human activities. To tackle the challenges, we first proposed the new diffusion-based multi-view exocentric encoder to extract the dense multi-scale exocentric features as the appearance conditions. Then, we introduced an Exo2Ego view translation prior to provide coarse yet spatially aligned egocentric features as a concatenation guidance. Finally, we inserted temporal layers into Ego Unet for improved temporal consistency across ego frames. Exo2Ego-V produced significant improvements on challenging Ego-Exo4D dataset [15] over SOTA approaches.

**Limitations and future work.** Exo2Ego-V focuses on Exo2Ego video generation on several categories of skilled human activities. It remains challenging but is worthwhile researching on more general activities. Exploring Gaussian Splatting as translation prior is also a promising direction.

## 6 Acknowledgment

This project is supported by the Mike Zheng Shou's Start-Up Grant from NUS. Jia-Wei Liu is also supported by NUS IDS-ISEP scholarship. Thanks to Zihang Xia for helpful discussions.

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

# A Supplemental material

## A.1 Implementation details

**Exo2Ego translation prior details.** Inspired by ReconFusion [56], our Exo2Ego translation prior $\Psi$ is based on a light PixelNeRF with 6-layer MLPs (Fig. 9) for higher efficiency together with a ResNet34 [17] pretrained on the ImageNet dataset to extract exo image features. For each synchronized timestep of the 4 exocentric videos and 1 egocentric video, we extract 4 exo frames and 1 ego frame and randomly sample 128 pixel rays from these 5 images and sample 3D points $\mathbf{x}$ for each iteration. Then, we add positional embedding to these points $\gamma(\mathbf{x})$ and query their latent features $\mathbf{f}(\mathbf{x})$ by projecting them on the latent images. Then we concatenate them with viewing direction as input to our translation prior as in Fig. 9. We conduct volumetric rendering at the third layer for the latent features, and then input the latent features to the last 3 layers for final pixel color and latent features.

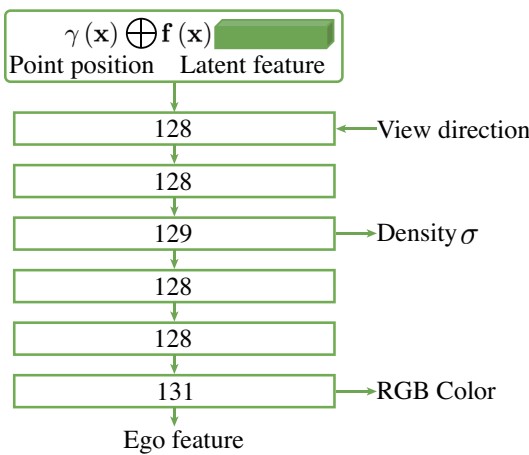

Figure 9: Network details for our Exo2Ego translation prior.

For egocentric video generation, we only sample rays from $32 \times 32$ ego frames so that the rendered ego features are spatially aligned with the noisy ego latent. Therefore, we concatenate the rendered ego features with the noisy ego latent as the input to our egocentric video diffusion pipeline.

**Egocentric camera unprojection.** The ego camera from Ego-Exo4D dataset [15] utilizes the FisheyeRadTanThinPrism (Fisheye624) model, which accounts for thin-prism distortion. This model includes four additional coefficients: $s_0$, $s_1$, $s_2$, $s_3$. The projection function is:

$$
\begin{aligned}
\mathbf{u} &= \mathbf{fx} * (\mathbf{u_r} + \mathbf{t_x}(\mathbf{u_r}, \mathbf{v_r}) + \mathbf{tp_x}(\mathbf{u_r}, \mathbf{v_r})) + \mathbf{c_x}, \\
\mathbf{v} &= \mathbf{fy} * (\mathbf{v_r} + \mathbf{t_x}(\mathbf{u_r}, \mathbf{v_r}) + \mathbf{tp_x}(\mathbf{u_r}, \mathbf{v_r})) + \mathbf{c_y}, \\
\mathbf{tp_x}(\mathbf{u_r}, \mathbf{v_r}) &= \mathbf{s_0}\mathbf{r}(\theta)^2 + \mathbf{s_1}\mathbf{r}(\theta)^4, \\
\mathbf{tp_y}(\mathbf{u_r}, \mathbf{v_r}) &= \mathbf{s_2}\mathbf{r}(\theta)^2 + \mathbf{s_3}\mathbf{r}(\theta)^4, \\
\mathbf{r}(\theta) &= \sqrt{(\mathbf{u} - \mathbf{c_x})^2/\mathbf{f_x^2} + (\mathbf{v} - \mathbf{c_y})^2/\mathbf{f_y^2}}, \\
\phi &= \arctan((\mathbf{u} - \mathbf{c_x})/\mathbf{f_x}, (\mathbf{v} - \mathbf{c_y})/\mathbf{f_y})
\end{aligned}
\tag{7}
$$

$\mathbf{u}, \mathbf{v}$ are the camera pixel coordinates and $\phi, \theta$ are the world point. $\mathbf{fx}, \mathbf{fy}$ are the focal lengths. Its parametrization contains 4 additional coefficients: $\mathbf{s_0}, \mathbf{s_1}, \mathbf{s_2}, \mathbf{s_3}$. Firstly we use the Newton method to calculate the $\mathbf{u_r}, \mathbf{v_r}$ and then we calculate $\phi, \theta$ using the above unprojection method. Finally, we sample 3D points along the calculated directions $\phi, \theta$.

## A.2 Training details

We optimize our Exo2Ego-V using Adam optimizer [24]. We set the learning rate of the multi-view exocentric encoder and the egocentric diffusion model as 0.00001, and we set the learning rate of view translation prior as 0.0001. We first train the translation prior with 500K iterations on a single A100 GPU for 36 hours, and then optimize our Exo2Ego spatial appearance translation with 500K iterations on 8 A100 GPUs for 48 hours, and finally finetune our temporal motion module with 100K iterations on 8 A100 GPUs for 40 hours, all using the PyTorch [40] deep learning framework.

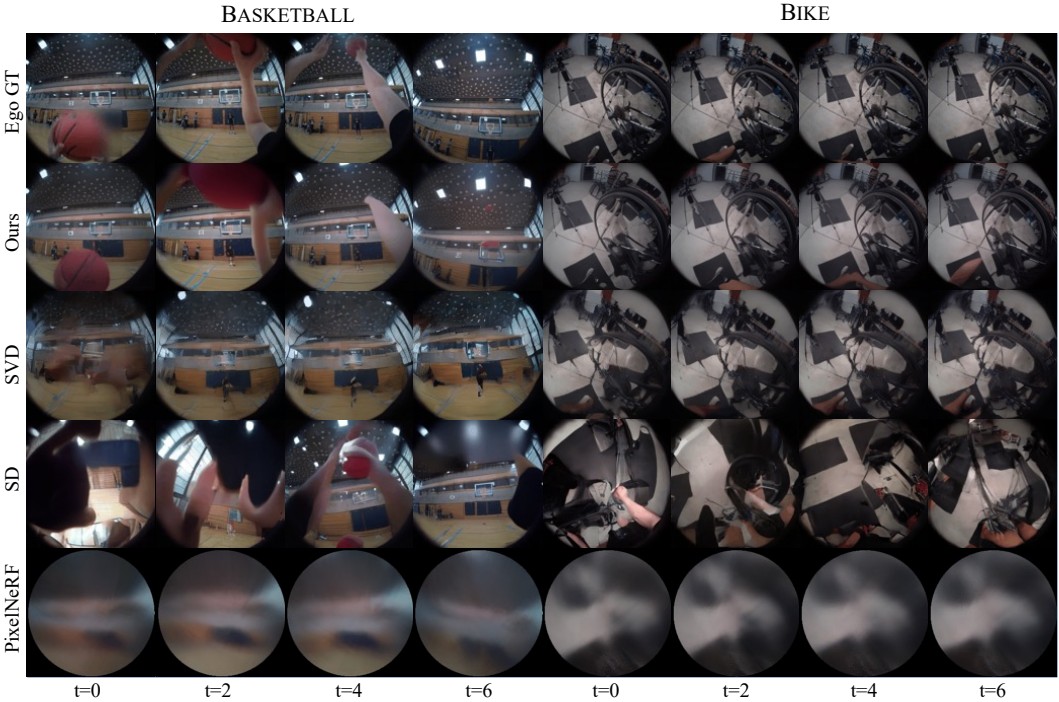

Figure 10: More qualitative comparisons of our method against SOTA approaches on **unseen actions**.

## A.3 Additional qualitative results

We provide more qualitative comparisons of Exo2Ego-V over SOTA approaches on unseen actions in Fig. 10. Our method achieves the best performance across all approaches.

## A.4 Feature Visualization

In Fig. 11, we present our ego feature visualization results using the Exo2Ego prior. The visualization results clearly represent the contents of the ego views. This indicates that our Exo2Ego prior can effectively extract and transmit the important information from ego views to the multi-view exocentric encoder.

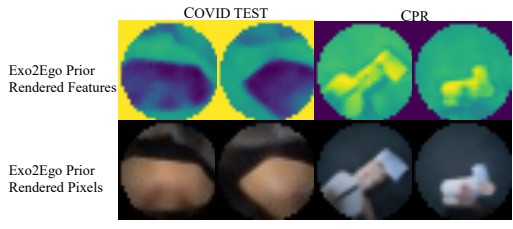

Figure 11: Exo2Ego prior feature visualization.

## A.5 Reasoning Efficiency

We provide the inference time comparison as shown in Tab. 5, where the inference time of our method to generate an 8-frame egocentric video is 9.06 second, which is comparable with other baselines. We believe it is feasible to use our model in offline applications to generate egocentric videos from the exocentric videos, such as capturing exocentric cooking videos and generating corresponding egocentric videos offline for cooking skills learning. Improving the inference speed towards real-time is very promising and we leave it as future works.

Table 5: Inference time of our method in comparison with baselines.

|  | Ours | SVD [5] | SD [46] | PixelNeRF [60] |
|---|---|---|---|---|
| Inference time (second) | 9.06 | 4.26 | 6.91 | 5.65 |

## A.6 Preliminary on denoising diffusion probabilistic models (DDPMs)

DDPMs [19] are generative frameworks designed to synthesize data by reproducing a consistent forward Markov chain $x_1, \ldots, x_T$. The process begins from a random noise distribution and progres-

sively denoise the noisy contents to the clean data. Considering a data distribution as $x_0 \sim q(x_0)$, the Markov transition $q(x_t|x_{t-1})$ is conceptualized as a Gaussian distribution by a variance $\beta_t \in (0, 1)$:

$$q(x_t|x_{t-1}) = \mathcal{N}(x_t; \sqrt{1 - \beta_t}x_{t-1}, \beta_t\mathbb{I}), \quad t = 1, \dots, T. \tag{8}$$

Under the Bayes and Markov principles, the conditional probabilities can be derived as:

$$q(x_t|x_0) = \mathcal{N}(x_t; \sqrt{\bar{\alpha}_t}x_0, (1 - \bar{\alpha}_t)\mathbb{I}), \, q(x_{t-1}|x_t, x_0) = \mathcal{N}(x_{t-1}; \tilde{\mu}_t(x_t, x_0), \tilde{\beta}_t\mathbb{I}), \, t = 1, \dots, T, \tag{9}$$

where $\alpha_t = 1 - \beta_t$, $\bar{\alpha}_t = \prod_{s=1}^{t} \alpha_s$, $\tilde{\beta}_t = \frac{1-\bar{\alpha}_{t-1}}{1-\bar{\alpha}_t}\beta_t$, $\tilde{\mu}_t(x_t, x_0) = \frac{\sqrt{\bar{\alpha}_t}\beta_t}{1-\bar{\alpha}_t}x_0 + \frac{\sqrt{\alpha_t}(1-\bar{\alpha}_{t-1})}{1-\bar{\alpha}_t}x_t$. DDPMs utilize a reverse approach to synthesize the chain $x_1, \dots, x_T$:

$$p_\theta(x_{t-1}|x_t) = \mathcal{N}(x_{t-1}; \mu_\theta(x_t, t), \Sigma_\theta(x_t, t)), \quad t = T, \dots, 1. \tag{10}$$

The model parameters $\theta$ are optimized to ensure the synthesized reverse sequence aligns with the forward sequence.

## A.7 Broader impacts

Our work can generate egocentric videos from exocentric videos. Since the scale of egocentric dataset is still much less than the exocentric dataset, our method has the potential to improve the egocentric vision such as egocentric perception by generating more egocentric data from the exocentric data. Our Exo2Ego-V also support applications on AI assistant and augmented reality by generating egocentric videos from exocentric videos. We believe our method will not bring negative societal impacts.

