# OpenReview forum: "Exocentric-to-Egocentric Video Generation"
_NeurIPS.cc/2024/Conference — NeurIPS 2024 poster_

### Official Review · Reviewer_aauy · 2024-07-03

**Soundness:** 3
**Presentation:** 4
**Contribution:** 3
**Rating:** 7
**Confidence:** 4

**Summary:**

This paper introduces a novel method for generating egocentric videos from exocentric videos of daily-life skilled human activities. This task is very challenging because of the significant viewpoint variations, sparsely posed exo videos, and dynamic environments. The authors propose to use a PixelNeRF-informed diffusion method to address the problem and achieves good visual performance and SOTA accuracy.

**Strengths:**

This paper addresses a challenging problem and achieves great performance. The proposed method is novel and reasonable that effectively leverages the camera poses and multi-view video features. The experiments show that it achieves the SOTA accuracy.

**Weaknesses:**

The proposed method lacks specific designs tailored for the "Exocentric-to-Egocentric" task. "exo" and "ego" perspectives are relative to "the person" in the video, yet the proposed method does not include any design elements focused on the person. The method actually has broad applicability to novel view video synthesis with significant viewpoint variations. One of the goals of the "Exocentric-to-Egocentric" task is to generate dynamic hand motions and object state changes. However, the visualization results in the paper show bad hand-object motions synthesis result. In summary, while the model excels at reconstructing the geometry of the surrounding environment, it lacks focus on hand-object dynamics which is important in egocentric video.

**Questions:**

Could you provide a failure case analysis? Do you think the fine-grained hand-object interactions causes most of the failure cases?

**Limitations:**

Limitation included

---

> ### Author Rebuttal · Authors · 2024-08-07
>
> **Thank you for recognizing our work and the valuable comments.**
>
> # W1: Person-related design.
>
> Thanks for your advice. We totally agree with you that person-related designs such as human hand pose prior or object states information would be very useful to improve the performance of Exo2Ego video generation. However, our method is the first work to achieve exocentric-to-egocentric video generation for daily-life real-world skilled human activities. For such challenging daily-life activities in real-world environments such as cooking in the kitchen, playing basketball on the court, **the exocentric cameras have to be placed far from the human** in order to capture the complete human activities. As a result, it is particularly difficult to capture detailed human hand-object regions on the exocentric cameras. **Therefore, it is particularly challenging to obtain human hand pose prior information or object state information from exocentric cameras from preprocessing**, and **generating human-object interactions is also one of the particular challenges for our Exo2Ego video generation task.**
>
> Therefore, **our method focuses on the general exocentric to egocentric video generation and also has broad applicability to novel view video synthesis with significant viewpoint variations as you suggested.** In addition, **our method achieves much better performances on the human-object regions compared to the baselines, as shown in Fig. 3 of rebuttal pdf.** Our method can accurately generate the human hands that are operating the CPR on the model, while SVD generates much worse results. We agree that adding more person-related designs can further improve the performance and leave this as faithful future directions.
>
> # Q1: Failure case analysis.
>
> Thanks for your advice. **We provide two failure cases on Fig. 7 of rebuttal pdf.** Yes, most of our failure cases are caused by the fine-grained human-object interactions, **due to the significant differences between exocentric and egocentric viewpoints, as well as far-away placed exocentric cameras in the daily-life environments.** Although we achieve much better performance than baselines, we believe there is room for improvements especially based on your advice of person-related designs. We leave this as faithful future directions.

---

> ### Author Response · Authors · 2024-08-13
> **Could you kindly review our response?**
>
> Dear Reviewer,
>
> Thank you once again for your feedback. As the rebuttal period is nearing its conclusion, could you kindly review our response to ensure it addresses your concerns? We appreciate your time and input.
>
> Best regards,
>
> Authors of 10425

---

### Official Review · Reviewer_qWdm · 2024-07-10

**Soundness:** 2
**Presentation:** 2
**Contribution:** 2
**Rating:** 5
**Confidence:** 4

**Summary:**

This paper proposes a novel method for generating egocentric videos from multi-view exocentric videos using diffusion-based techniques. This method addresses the challenges of viewpoint variation and dynamic motion by employing a multi-view exocentric encoder and a view translation prior, along with temporal attention layers to enhance temporal consistency. Experiments results on the Ego-Exo4D dataset verify the effectiveness of the proposed method.

**Strengths:**

1.	The logic of the paper is reasonable.

2.	The motivation is interesting.

**Weaknesses:**

1.	The exocentric-to-egocentric view translation prior is very similar to the existing work ReconFusion, and the authors need to clarify the differences.

2.	In the egocentric video generation pipeline, why perform spatial attention before temporal attention instead of following the TimeSformer? Are there relevant experiments for further clarification?

3.	The reasoning efficiency of the whole process should be evaluated. This determines whether it can be practically applied.

4.	How does the number of exocentric videos affect performance? In real scenarios, it is difficult to capture 4 time-synchronized exocentric videos at the same time. Besides, do the 4 exocentric videos have to be evenly distributed around the scene?

5.	The evaluation dataset is so single that it is impossible to validate the generalizability of the method on more diverse data. Furthermore, Ego-Exo4D contains a wide range of human activities, why only 5 of these categories are selected for the experiment?

6.	The baselines compared are too few to adequately validate the superiority of the method.

7.	Writing needs further improvement, tenses should be correct (Line 185, Page 5) and contextual expressions need to be consistent (Unet or UNet).

**Questions:**

After reviewing the response letter, some of my concerns have been partially addressed. However, the paper's novelty, primarily inspired by ReconFusion, appears to have a weak technical contribution. Additionally, the evaluation is limited by the dataset used. Consequently, I have adjusted my score to a the borderline reject.

**Limitations:**

yes

---

> ### Author Rebuttal · Authors · 2024-08-07
>
> **Thanks for helpful comments.**
> # W1: Differences with ReconFusion.
> Our Exo2Ego prior is inspired by ReconFusion and based on PixelNeRF (see L199-L200, L204-L206 of main paper), but it significantly differs from ReconFusion in following aspects.
> 1) **The task of our method is significantly different from ReconFusion.** ReconFusion targets 3D static scene reconstruction, and thus it trains a diffusion prior on 3D static datasets and explores how to distill their diffusion prior into a 3D model through regularization losses. In contrast, our method is the first work to approach the challenging exocentric to egocentric dynamic video generation for skilled human activities with significant differences between exocentric and egocentric viewpoints.
> 2) **The design of our Exo2Ego prior is different from ReconFusion’s diffusion prior.** ReconFusion directly trains a diffusion prior that includes a PixelNeRF and diffusion UNet, and then explores distilling from the pretrained diffusion prior. In contrast, our method first trains a PixelNeRF-based Exo2Ego prior, and then we integrate this Exo2Ego prior into our complete Exo2Ego video generation pipeline and finetune Exo2Ego prior to provide the coarse yet spatially aligned egocentric features for our egocentric video generation. In addition, the PixelNeRF from the diffusion prior of ReconFusion is a 2-layer MLP, while our Exo2Ego prior is a 6-layer MLP.
> 3) **We further propose the multi-view exocentric encoder to provide fine-grained multi-scale exocentric features for egocentric video generation. Our multi-view exocentric encoder and Exo2Ego prior complement with each other.** Our complete Exo2Ego video generation model is the first video-level model for Exo2Ego video generation, and achieves new SOTA.
> # W2: Ablation on temporal-spatial.
> We conduct an ablation to first perform temporal attention and then spatial attention for our model. As shown in the following table, the original spatial-temporal model is slightly better than the temporal-spatial model in terms of PSNR and SSIM, and slightly worse for LPIPS. Since previous video generation methods such as AnimateDiff [16], SVD [5] all follow the spatial-temporal pipeline, our method also follows such spatial-temporal attentions. It is noted that since our temporal layers are finetuned from AnimateDiff [16] that is a spatial-temporal pipeline, our spatial-temporal model performs slightly better than temporal-spatial pipeline. **We provide qualitative results on Fig. 4 of rebuttal pdf where ours generates more accurate human-objects.**
> ||Unseen Action|||Unseen Take|||
> |-|-|-|-|-|-|-|
> || PSNR↑| SSIM↑| LPIPS↓| PSNR↑| SSIM↑| LPIPS↓|
> |Spatial-Temporal| **17.37**| **0.493**|0.408|**17.71**|**0.504**|0.456|
> |Temporal-Spatial| 17.00| 0.484|**0.402**|17.29|0.490|**0.443**|
> # W3: Reasoning efficiency.
> Please refer to common response 2.
> # W4: Number of exocentric videos.
> **We conduct additional ablations on the number of exocentric views of 4, 3, 2, 1 views on the following table**. Our method’s performance slightly drops with the reduction of exocentric views. We provide qualitative comparisons on Fig. 5 of rebuttal pdf. **Even with only one exocentric view, our performance is still much better than other baselines, which significantly demonstrates the effectiveness of our approach with various numbers of exocentric views. This avoids the necessity of capturing time-synchronized exocentric videos at the same time, making our method more applicable for real-world applications**.
> ||Unseen Action|||Unseen Take|||
> |-|-|-|-|-|-|-|
> ||PSNR↑|SSIM↑|LPIPS↓|PSNR↑|SSIM↑|LPIPS↓|
> |Ours w/ 4 Views|**17.37** |**0.493**|0.408|**17.71**|**0.504**|0.456|
> |Ours w/ 3 Views|17.23|0.486|**0.383**|17.40|0.489|**0.428**|
> |Ours w/ 2 Views|16.93|0.474|0.399|17.02|0.479|0.445|
> |Ours w/ 1 View|17.02|0.478|0.395|17.20|0.476|0.439|
> # W5: Evaluation dataset.
> We selected Ego-Exo4D dataset [15] as our main evaluation dataset due to its significant challenges of the complexity and diversity of daily-life scenarios, large differences between exo and ego viewpoints, complex human-object interactions, and its largest Ego-Exo data scale with 1286 hours of video. **Ego-Exo4D has 8 domains: Cooking, Health, Bike Repair, Music, Basketball, Rock Climbing, Soccer, Dance. It is different from Ego4D and does not contain a wide range of activities**. Since our task focuses on Exo2Ego video generation, **We select 5 categories that significantly emphasize both egocentric and exocentric activities, such as human-object interactions in both Ego and Exo viewpoints (see L235-L238 of main paper)**. Others like Dance, Soccer, Rock Climbing are mainly exocentric activities and contain very few human or human-object interactions in the egocentric view, which weakens the usefulness of generating egocentric views and thus are not included in our experiments.
>
> **Our method can be generalized to other EgoExo datasets. Please refer to common response 1 for more results on H2O dataset.**
> # W6: Baselines.
> Our method is the first work to achieve Exo2Ego video generation for daily-life real-world skilled human activities. **This task is particularly challenging and there are no available baselines to compare with.** Therefore, we design three baselines including the **SOTA open-sourced video generation model** SVD, the **image generation method** SD, and the **3D-based method** PixelNeRF. We modify the input and condition modules of these baselines and train and evaluate them on the Ego-Exo4D dataset. We selected SVD because it is the SOTA open-sourced video generation model. **Therefore, we believe our baselines are adequate to demonstrate the superiority of our method. We additionally inference two more pre-trained image-to-video and video-to-video generation models on the Ego-Exo4D dataset as shown in Fig. 6 of rebuttal pdf, where these methods totally fail to generate egocentric videos from exocentric videos input**.
> # W7: Writing.
> We have corrected such typos.

---

> ### Author Response · Authors · 2024-08-12
> **Response to Reviewer qWdm (1/2)**
>
> Dear Reviewer qWdm,
>
> Thank you for updating your comment in the Reviews Questions. We are glad that we have addressed some of your concerns. Here we further respond to your remaining concerns about the paper novelty and evaluation dataset.
>
> # Paper novelty.
>
> **We respectfully disagree with your statement that our paper’s novelty is primarily inspired by ReconFusion.** We provided detailed responses about the differences between our Exo2Ego prior with ReconFusion in our rebuttal to address your concern about weakness 1. **Here we further clarify the overall novelty and technical contributions of our paper in comparison with ReconFusion.**
>
> 1. **Our method is the first work to achieve exocentric-to-egocentric video generation for daily-life real-world skilled human activities.** This task is particularly challenging in terms of the significant differences between the exocentric and egocentric viewpoints, as well as complex human-object interactions and real-world environments. In contrast, ReconFusion focuses on traditional 3D static scene reconstruction. **Therefore, our method is fundamentally different from ReconFusion.**
>
> 2. **To address the above challenges of Exo2Ego video generation, we propose a new diffusion-based multi-view exocentric encoder to extract fine-grained multi-scale exocentric features, as well as an Exo2Ego view translation prior to render spatially aligned geocentric features.** These two modules complement each other with fine-grained exocentric features and spatially aligned egocentric features to provide conditional information for **our egocentric video generation pipeline.** Our complete Exo2Ego video generation model is **the first video-level model for Exo2Ego video generation, and significantly outperforms previous approaches by a large margin of 35% for LPIPS.**
>
> 3. **Therefore, the technical contributions of our method are three-folds: 1) Diffusion-based multi-view exocentric encoder, 2) Exo2Ego view translation prior, 3) The first Exo2Ego video generation pipeline.** We conducted extensive ablation studies to demonstrate the effectiveness of each component in our main paper.
>
> As discussed in our main paper (L199-L200, L204-L206) and rebuttal W1, **only the Exo2Ego view translation prior is partially inspired by ReconFusion — both our Exo2Ego prior and ReconFusion are based on PixelNeRF, but our Exo2Ego view translation prior is significantly different from ReconFusion. Furthermore, our proposed first exocentric-to-egocentric video generation model, our multi-view exocentric encoder, and our challenging Exo2Ego video generation task definition are all fundamentally different from ReconFusion.** **Below we further discuss the detailed technical differences between our Exo2Ego translation prior with ReconFusion.**
>
> 1. **The design and functionality of our Exo2Ego prior is different from ReconFusion’s diffusion prior.** Our method first trains a PixelNeRF-based Exo2Ego prior, and then we integrate this Exo2Ego prior into our complete Exo2Ego video generation pipeline and finetune Exo2Ego prior to provide the coarse yet spatially aligned egocentric features for our egocentric video generation. In contrast, ReconFusion directly trains a diffusion prior that includes a PixelNeRF and diffusion UNet, and then explores distilling from the pretrained diffusion prior.
>
> 2. **Our Exo2Ego prior is specially designed for rendering egocentric views from exocentric inputs.** To achieve this, we differentiate the egocentric views and exocentric views for our Exo2Ego prior model – it only extracts exocentric views’ ResNet features as condition signals to render the pixel colors and features. **Our exocentric and egocentric viewpoints are sparse and significantly different from each other. In contrast, ReconFusion does not differentiate between views and is trained on dense multi-view 3D static scenes datasets that contain hundreds of views for each 3D static scene.**
>
> 3. **Our Exo2Ego prior’s model architecture is different from ReconFusion.** We employ a 6-layer MLP as our Exo2Ego prior base model to handle the complex Exo2Ego translation with large Exo-Ego viewpoint differences, while ReconFusion only adopts a 2-layer MLP.
>
> 4. **Our Exo2Ego prior’s training strategy is different from ReconFusion.** We first pretrain our Exo2Ego prior on the Ego-Exo4D dataset. Then, we integrate our Exo2Ego prior into our Exo2Ego video generation pipeline and we alternatively finetune Exo2Ego prior and the complete pipeline during training. This strategy ensures that Exo2Ego prior renders spatially-aligned feature maps necessary for egocentric video generation, as well as keeps the original scene reconstruction capability.

---

> ### Author Response · Authors · 2024-08-12
> **Response to Reviewer qWdm (2/2)**
>
> # Evaluation dataset.
>
>
> We explained in our **rebuttal W5** that we selected Ego-Exo4D dataset [15] as our main evaluation dataset due to its **significant challenges of the complexity and diversity of daily-life scenarios, large differences between exo and ego viewpoints, complex human-object interactions, and its largest Ego-Exo data scale with 1286 hours of video.** We selected the 5 out of 8 categories from the Ego-Exo4D dataset that significantly emphasize both egocentric and exocentric activities, such as human-object interactions in both Ego and Exo viewpoints.
>
> **We have demonstrated that our method can be generalized to other EgoExo datasets such as H2O dataset in common response 1**, where our method achieves much better performance than SVD, with significant 30.3% improvement over SVD in terms of LPIPS. **We are glad that Reviewer ANQv, who suggested this H2O dataset evaluation, has responded that our rebuttal has solved all Reviewer ANQv’s concerns, and we will add these results to provide more insights to readers.**
>
>
> **Therefore, we believe our dataset evaluation is very challenging and extensive to evaluate the superiority of our proposed method.**

---

> ### Author Response · Authors · 2024-08-14
> **Could you kindly let us know whether we address all your concerns?**
>
> Dear Reviewer,
>
> Thank you once again for your feedback. We provided additional responses towards your remaining concerns. As the rebuttal period is nearing its conclusion, could you kindly review our response to ensure it addresses your concerns? We appreciate your time and input.
>
> Best regards,
>
> Authors of 10425

---

### Official Review · Reviewer_ANQv · 2024-07-12

**Soundness:** 3
**Presentation:** 3
**Contribution:** 3
**Rating:** 6
**Confidence:** 4

**Summary:**

This paper proposes a novel diffusion-based video generation method that translates exocentric views to egocentric view. Overall, I think the idea is novel and the method performs well on multiple daily human activities.

**Strengths:**

1.	The application of view translation using Nerf-based approach seems interesting, despite minor improvement on several metrics.
2.	Multiview exocentric encoder and temporal modeling is proven effective in supporting view transfer.
3.	The paper is well written and easy to follow.

**Weaknesses:**

1.	In Line 157-159, the authors mentioned that encoding multiview exocentric videos using CLIP image features lacks fine-grained details. However, there is no sufficient evidence in the experiments. For example, comparing the proposed method with global CLIP features or intermediate spatial CLIP features as conditions.
2.	Generally, as most of the takes in EgoExo4d contain 4 exocentric views, which are significantly fewer than dozens of views used in 3D reconstruction approaches. The effectiveness of reconstructing a 3D scene using such sparse viewpoints is questionable, not to mention the significant camera pose differences between egocentric and exocentric views.
3.	In the experiments, the rendered pixels of PixelNerf are extremely blurry, and thus it is not very convincing to use the CLIP features of rendered egocentric image in Eq.(4). To prove the effectiveness of exo2ego prior, the authors are encouraged to show some visualization results of the rendered egocentric features using PixelNerf.

**Questions:**

1. The proposed method outperforms the baseline SVD that performs video generation given the first frame as condition. It is not clear to me whether the authors also adopted first frame as the condition while training the model since the first frame is a very strong prior. If not, could the authors explain why the proposed method can outperform SVD without using such strong prior, when both methods are given the same exocentric views as conditions.
2.  It is not clear whether the proposed method can be generalized to other egoexo datasets that also contain camera poses, e.g. H2O [1], Assembly101[2].

[1] Kwon et al. H2O: Two Hands Manipulating Objects for First Person Interaction Recognition. ICCV21.

[2] Sener, Fadime, et al. Assembly101: A large-scale multi-view video dataset for understanding procedural activities.CVPR 2022.

**Limitations:**

Please refer to Weaknesses and Questions.

---

> ### Author Rebuttal · Authors · 2024-08-07
>
> **Thank you for recognizing our work and the valuable comments.**
>
> # W1: Ablation on Exocentric CLIP features.
> Thanks for your advice. We conduct additional ablation study by replacing our exocentric feature encoders with the CLIP exocentric features. As shown in the following table, **our method achieves much better performance compared to using CLIP features across all metrics on the unseen action and unseen take evaluations.** This clearly demonstrates the superiority of our multiview exocentric encoder in maintaining fine-grained details compared to the CLIP features. We also provide **qualitative ablation comparison on Fig. 4 of rebuttal pdf**, where our method with Exo encoder outperforms the one with CLIP features on many detailed regions such as left and right human arm and objects. We will add this ablation result to our final paper.
>
> |                    | Unseen Action |       |       | Unseen Take |       |       |
> |--------------------|---------------|-------|-------|-------------|-------|-------|
> |                    | PSNR↑          | SSIM↑  | LPIPS↓ | PSNR↑        | SSIM↑  | LPIPS↓ |
> | Ours w/ Exo Encoder | **17.37**    | **0.493** | **0.408** | **17.71** | **0.504** | **0.456** |
> | Ours w/ Exo CLIP    | 16.54         | 0.456 | 0.425 | 16.41       | 0.445 | 0.480 |
>
> # W2: Only 4 exocentric views are very challenging.
> The 4 sparse views and significant differences between exocentric and egocentric cameras are **exactly the key challenges for our Exo2Ego video generation task**. Therefore, **previous 3D reconstruction methods such as PixelNeRF fail in this task**, as shown in the baseline PixelNeRF results in Fig. 3 of main paper. To tackle these challenges, we propose our complete Exo2Ego-V pipeline with both multi-view exocentric encoder to extract multi-scale exocentric features and Exo2Ego translation prior to render coarse yet spatially aligned egocentric feature maps. **These two designs complement each other: the exocentric features are fine-grained but not spatially aligned with the ego input, while the egocentric features are a bit coarse but they are spatially aligned with the ego input.** Our experiments demonstrate that our Exo2Ego-V pipeline achieves much better performance than previous reconstruction and generation baselines for such a challenging task with large Exo-Ego viewpoint differences.
>
> # W3: Visualization of rendered egocentric features.
> As shown in Fig. 6 of appendix, our Exo2Ego prior rendered both the egocentric features and images. We want to clarify that we utilize two types of egocentric features: the egocentric features rendered directly from the Exo2Ego prior, and the CLIP features of the egocentric images rendered from the Exo2Ego prior. The first one is spatially aligned with the egocentric input latent and is directly concatenated with the input to provide spatial guidance, and the latter is utilized as the cross attention computation to maintain the original diffusion architecture.
>
> Following your suggestino, we provide more **visualizations of rendered egocentric images and rendered egocentric features on Fig. 2 of rebuttal pdf**. As shown in the figure, **the rendered feature maps can accurately encode the geometric information such as the desktop shape, and the CPR models**. Our rendered features are spatially aligned with the egocentric inputs and are complemented with the fine-grained exocentric features extracted from our exocentric encoder. Although the rendered feature maps could be coarse for complex scenarios, but they are spatially aligned egocentric features (as mentioned in L150-L151 of main paper) can provide spatial guidance for our egocentric generation pipeline,  and our ablation studies also demonstrate the effectiveness of our proposed Exo2Ego prior.
>
> **We conducted an ablation study on our proposed Exo2Ego translation prior in Tab. 2 and Fig. 5 of the main paper, where removing Exo2Ego prior results in worse performance for PSNR and SSIM**. In addition, **through the visualization of Fig. 5, removing Exo2Ego prior (second row, second column) results in the missing of right human arm.** These ablations **demonstrate the effectiveness of our Exo2Ego prior.**
>
> # Q1: SVD training.
> **For fair comparisons, we provide the same 4 exocentric videos as inputs for our method and all baseline methods including SVD.** Therefore, **we modify the condition blocks of SVD to take 4 exocentric videos as input and finetune the pretrained SVD model on the Ego-Exo4D dataset.** As such, all methods are given the same exocentric videos as inputs to generate corresponding egocentric videos, and they are all trained and tested on the same dataset to achieve fair comparisons among each other.
>
> # Q2: Generalization on other EgoExo datasets.
>
> Thanks for your advice! Our method can be generalized to other EgoExo datasets. Following your suggestion, **we additionally train and evaluate our method on the H2O dataset, and compare our method with the SOTA baseline - SVD on the following table**. As shown in the following table, **our method achieves much better performance than SVD, with significant 30.3% improvement over SVD in terms of LPIPS**. **We provide qualitative comparisons of our method against SVD on Fig. 1 of rebuttal pdf**, where **our method achieves much more photorealistic and accurate results than SVD**, such as the details of human hands, interacted objects, and the environments. **These can demonstrate our method can be generalized to other datasets**, and we will add this result on the final paper.
>
> |      | PSNR↑      | SSIM↑      | LPIPS↓     |
> |------|-----------|-----------|-----------|
> | Ours | **18.60** | **0.581** | **0.189** |
> | SVD  | 16.53     | 0.468     | 0.271     |

---

> > ### Comment · Reviewer_ANQv · 2024-08-10
> > **Official Comment by Reviewer ANQv**
> >
> > Dear authors,
> >
> > Thanks for your responses and explanations. Overall, the rebuttal solves all my concerns. With these additional experiments, this paper would provide more insights to readers.
> >
> >
> > Regarding the response to W3 (visualization of rendered egocentric features), I have a few more questions that do not affect the rating.
> >
> >
> > The visualization results in Figure 2 show that the rendered feature maps/pixels generally capture the low-frequency of the visual content, e.g. table. , and these maps/pixels have exact spatial correspondence to the GT ego frame as mentioned by authors.
> >
> >
> > Given the GT ego frame/rendered feature map/rendered rgb image, is it possible to map one pixel (e.g. center of the table) back to each of the four exocentric frames given the ego and exo camera poses? Could you briefly introduce the solution if possible?
> >
> >
> > If the mapping is available, this could provide more fine-grained understandings of how each pixel in the rendered feature map/rgb image leverages the pixel-wise exocentric information from each exo camera. I think this is useful when the field-of-ego-view is under occlusion in one/two exocentric cameras, and we would like to know which exo camera(s) provide valuable visual clues to support generation in the ego view.

---

> ### Author Response · Authors · 2024-08-11
> **Response to Reviewer ANQv**
>
> Dear Reviewer ANQv,
>
> **Thanks for your valuable comments! We are glad that our rebuttal solves all your concerns! We will add these additional experiments to the final paper to provide more insights to readers!**
>
> Regarding your additional question, “Given the GT ego frame/rendered feature map/rendered rgb image, is it possible to map one pixel (e.g. center of the table) back to each of the four exocentric frames given the ego and exo camera poses?”
>
> **Yes, it is possible to map one pixel from the ego frame back to each of the four exocentric frames given the ego and exo camera poses.** To achieve that, we also require egocentric depth information $\mathbf{D}$ which can be conveniently obtained by additionally rendering the sampled points depths with their density values $\sigma$ along each ray using our Exo2Ego prior.
>
> Specifically, we can first backproject one egocentric pixel $\left(u, v\right)$ to the 3D space, denoted as $\mathbf{X}$, using egocentric cameras pose (to provide projection direction) together with the egocentric depth at that pixel (to provide the projection distance) as follows,
>
> $\mathbf{X}=E_{\mathrm{ego}}^{-1}\cdot K_{\mathrm{ego}}^{-1}\cdot\mathbf{D}\left[u, v\right]\cdot\left[u, v, 1\right]^{\mathrm{T}}$,
>
> where $E_{\mathrm{ego}}$ and $K_{\mathrm{ego}}$ are the extrinsics and intrinsics of egocentric camera, and $\mathbf{D}$ is the depth map for egocentric frame.
>
> After that, we can project $\mathbf{X}$ to each of the four exocentric frames using the exocentric camera poses $E_{\mathrm{exo}}$ and $K_{\mathrm{exo}}$.
>
> $\left(u_{\mathrm{exo}}, v_{\mathrm{exo}}, 1\right)=K_{\mathrm{exo}}\cdot E_{\mathrm{exo}}\cdot\mathbf{X}$,
>
> where $\left(u_{\mathrm{exo}}, v_{\mathrm{exo}}\right)$ are the projected exocentric pixels.
>
> As you suggested, this could provide more fine-grained understandings of how each pixel in the rendered feature map/rgb image leverages the pixel-wise exocentric information from each exo camera.
>
> Thank you very much for recognizing our work and valuable comments!
>
> Best regards,
>
> Authors of 10425

---

### Official Review · Reviewer_W4kZ · 2024-07-13

**Soundness:** 3
**Presentation:** 3
**Contribution:** 3
**Rating:** 6
**Confidence:** 4

**Summary:**

This paper deals with the task of exocentric-to-egocentric video generation. It presents a diffusion-based framework of exo2ego-v to tackle the challenges of the significant variations between exocentric and egocentric viewpoints and high complexity of dynamic motions and real-world daily-life environments. It propose a multi-view exocentric encoder to extract the multi-scale multi-view exocentric features as the appearance conditions for egocentric video generation. It also designs an exocentric-to-egocentric view translation prior based on PixelNeRF to provide coarse yet spatially aligned egocentric features as a concatenation guidance for egocentric video generation. Experiments on a part of Ego-Exo4D dataset show the effectiveness of the proposed method.

**Strengths:**

1) The paper presents a diffusion-based framework to first address the challenging task of exocentric-to-egocentric video generation for real-world skilled human activities.
2) To tackle the challenges of exocentric-to-egocentric video generation, the paper proposes a new diffusion-based multi-view exocentric encoder and an Exo2Ego view translation prior that can extract dense exocentric features and spatially aligned egocentric features as conditions for the egocentric video diffusion pipeline.
3) Experiments on on 5 categories of skilled activities on the Ego-Exo4D dataset show that the proposed framework achieves best performance most evaluation cases.

**Weaknesses:**

1) The presentation of the paper needs further improvement.
    1.1 For the method part, the network architecture is not clear for the exocentric encoder and the egocentric video diffusion model. As a result, the feature dimension of many notations such as $F_{exo}$, $Z^{t}_{exo}$ are not known. It is also difficult to understand the procedure within Equation (1) and (4).
    1.2 For the experiment part, the qualitative results have taken too much space (e.g., Figure 3). Space should be saved for the presentation of the results on the unseen scenes (which are moved to appendix).
    1.3 If the test scenes are not seen during training, Exo2Ego translation prior (i.e., pixelNeRF) need be trained for the new scenes. During testing, egocentric video frames are not available, while it is said in the paper that four exocentric frames and one egocentric frame are used to train the pixelNeRF.
2) As shown by the results on the more challenging unseen scenes (Table 3), the proposed method sometimes gets worse performance than the baseline methods. The authors should give more analysis and discussion for such results.

**Questions:**

1) Do you train one Exo2Ego prior model for each video, or for each synchronized timestep of a video? It is not clear for me how regular the Exo2Ego prior model need be retrained.
2) What is the dimension of the encoded features for multi-view exocentric videos, and how are they combined with the camera poses which have different dimensions?
3) How are the egocentric camera poses determined during testing? Do you directly use the poses provided by the dataset?
4) In Figure 5, the same results of EgoGT are shown twice. Please have a check.
5) Current evaluation metrics are averaged over the generated video. I wonder how the performance would change over time.

**Limitations:**

The paper can also show the runtime of the video generation framework.

---

> ### Author Rebuttal · Authors · 2024-08-07
>
> **Thank you for recognizing our work and the valuable comments.**
> # W1.1: Network architecture.
> The base network architecture of the exocentric encoder and the spatial modules of the egocentric video diffusion model are extensions of the Stable Diffusion [44]. Therefore, they consist of **four downsample layers, one middle layer, and four upsample layers, where each layer includes 2D convolution, self-attention, and cross-attention**. Since Stable Diffusion operates in the latent space, the feature dimension of input encoded noisy latents is $\mathbb{R}^{N\cdot F\times C\times\frac{H}{8}\times\frac{W}{8}}$.
>
> The exocentric attention features’ dimensions are different for different downsample, middle, and upsample layers with channel dimension varying at [320, 640, 1280], and the spatial dimensions $\frac{H}{8}\times\frac{W}{8}$ are downsampled by 2x2 or upsampled by 2x2 for each downsample and upsample layers, respectively. Eq. (1) and (4) are the **denoising processes of diffusion models** that predict the added noise of input noisy latents, except that in Eq. (1) we extract the exocentric features from each denoising layer. We will add these details to the main paper.
> # W1.2: Appendix results.
> Thanks for your advice. We will move the unseen scenes results to the main paper and move some qualitative results to the appendix.
> # W1.3, Q1: Exo2Ego prior training.
> **We only train one single generalizable Exo2Ego prior model for all exocentric videos and egocentric videos from the training set.** **For each training iteration, we randomly extract synchronized 4 exocentric frames and 1 egocentric frame to supervise our Exo2Ego prior (L197-L200).** During the complete training, each iteration will load different Exo-Ego videos from the training set to train our single generalizable Exo2Ego prior. Therefore, we only need to train one single Exo2Ego prior from all videos from the training set.
>
> During testing, our Exo2Ego translation prior can **generalize to test scenes that are not seen during training without retraining.** Therefore, we do not need to train the Exo2Ego translation prior for novel test scenes. As shown in our experiments, we evaluate our Exo2Ego-V on unseen actions, unseen takes, and unseen scenes without retraining our Exo2Ego prior on any of the test sets, which is very challenging. We will explain it clearer in our final paper.
> # W2: Unseen scenes results.
> The evaluation of the unseen scenes is particularly challenging for two reasons. 1) **We only train one single Exo2Ego-V model**, including the Exo2Ego prior, from the training dataset and evaluate its performance on the proposed 3 challenging test setups without any retraining or finetuning. 2) **The number of different scenes from Ego-Exo4D is relatively limited, but the geographic diversity of the captured scenes is particularly significant across the world**. Therefore, all methods perform worse on the unseen scene evaluation.
>
> **Despite these challenges, our method still outperforms the baseline methods in most metrics for the unseen scene evaluation.** As shown in Tab. 3 of the main paper, PixelNeRF only achieves the best PSNR metric for the Covid Test category, but it generates very blur results as shown in Fig. 7 since PSNR favors blurry images [37]. SVD occasionally achieves better metrics results but it only generates wrong scenes without any human-object interactions, as shown in Fig. 7 of main paper. In comparison, **our method is the only method that tries to generate both scenes and human-object interactions, such as the background and human arms and basketball in Fig. 7**. Although the performance on unseen scenes is not yet optimal, we believe jointly training our method with a more diverse multi-view dataset has the potential to improve performance and we leave this as future work.
> # Q2: Exocentric feature dimension.
> The exocentric attention features’ dimensions are different for different downsample, middle, and upsample layers with channel dimension varying at [320, 640, 1280], and the spatial dimensions $\frac{H}{8}\times\frac{W}{8}$ are downsampled by 2x2 or upsampled by 2x2 for downsample and upsample layers.
>
> As explained in L183-L187, to encode the relative camera poses information, we do not directly concatenate the camera poses with exocentric features. Instead, **we first encode the camera pose to 1280 dimension with a 2-layer MLP, and then add it to the denoising timestep embedding as the residuals for the noisy latents** in the following denoising blocks. In these following denoising blocks, there are additional MLP layers to map the channel dimension of the timestep and camera embedding to the same dimension as the corresponding noisy latents, so that these embeddings are added to the noisy latents as residuals.
> # Q3: Egocentric camera poses.
> Yes, we directly use the camera poses provided by the Ego-Exo4D dataset.
> # Q4: EgoGT Results.
> Thanks for your advice, we will remove the repeated EgoGT which was intended for symmetrical layout.
> # Q5: Performance over time.
> Following your suggestion, we compute the PSNR for each frame of the generated videos over time, and then average these metrics across all test videos for each time step respectively. We evaluate this on the three test sets of the Cooking category and compute all metrics. We report PSNR due to space limits, and SSIM and LPIPS show similar results. As shown in following table, **the performance remains relatively stable over time, and the intermediate frames perform slightly better than the final frames due to larger motion in the latter.**
>
> *Tab. 1 Quantitative metrics (PSNR) over time for Cooking category.*
> ||Frame 1|Frame 2|Frame 3|Frame 4|Frame 5|Frame 6|Frame 7|Frame 8|
> |-|-|-|-|-|-|-|-|-|
> |Unseen Action|17.35|17.42|17.44|17.44|17.43|17.35|17.38|17.27|
> |Unseen Take|17.63|17.78|17.81|17.76|17.79|17.86|17.71|17.60|
> |Unseen Scene|14.01|14.07|14.07|14.02|13.95|13.88|13.80|13.69|
> # L1: Inference time.
> Please refer to common response 2.

---

> > ### Comment · Reviewer_W4kZ · 2024-08-14
> >
> > Thanks for the authors' detailed responses which have addressed my initial concerns. After reading all the responses and comments from other reviewers, I would like to increase the rating to "Weak Accept".

---

> ### Author Response · Authors · 2024-08-13
> **Could you kindly review our response?**
>
> Dear Reviewer,
>
> Thank you once again for your feedback. As the rebuttal period is nearing its conclusion, could you kindly review our response to ensure it addresses your concerns? We appreciate your time and input.
>
> Best regards,
>
> Authors of 10425

---

### Author Rebuttal · Authors · 2024-08-07

We thank all the reviewers for their insightful and valuable comments. We also appreciate that the core contributions and the quality of our results are recognized in the review:

1.  **First work** to address the **challenging** task of Exo2Ego video generation. **New** diffusion-based multi-view exocentric encoder and an Exo2Ego view translation prior. **Best performance** on most evaluation cases. (Reviewer W4kZ)

2.  The application of view translation using the Nerf-based approach seems **interesting**. Multiview exocentric encoder and temporal modeling is proven **effective** in supporting view transfer. The paper is **well written and easy to follow**. (Reviewer ANQv)

3.  The task is **very challenging**. The proposed method is **novel and reasonable** that effectively leverages the camera poses and multi-view video features. Achieves **SOTA accuracy**. (Reviewer aauy)

4.  **The motivation is interesting**, and the logic of the paper is **reasonable**. (Reviewer qWdm)




**We have included additional figures in the global rebuttal PDF. Please refer to the PDF for further qualitative results.**

Below we first address the common advice from Reviewer ANQv and Reviewer qWdm to demonstrate the generalization of our method on other EgoExo datasets, as well as the common advice from Reviewer W4kZ and Reviewer qWdm on the inference time comparison.

For other concerns and comments, please refer to individual responses to each reviewer.



# Common 1: Generalization on other EgoExo datasets (Reviewer ANQv, qWdm).



Thanks for your advice. Our method can be generalized to other EgoExo datasets. Following your suggestion, **we additionally train and evaluate our method on the H2O dataset [1], and compare our method with the SOTA baseline - SVD on the following table**. As shown in the following table, **our method achieves much better performance than SVD, with significant 30.3% improvement over SVD in terms of LPIPS**. **We provide qualitative comparisons of our method against SVD on H2O dataset [1] on Fig. 1 of rebuttal pdf**, where **our method achieves much more photorealistic and accurate results than SVD**, such as the details of human hands, interacted objects, and the environments. **These can demonstrate our method can be generalized to other datasets**, and we will add this result on the final paper.

*Tab. 1 Quantitative comparison of our method against SVD on H2O dataset [1].*

|      | PSNR↑      | SSIM↑      | LPIPS↓     |
|------|-----------|-----------|-----------|
| Ours | **18.60** | **0.581** | **0.189** |
| SVD  | 16.53     | 0.468     | 0.271     |



# Common 2: Reasoning efficiency (Reviewer W4kZ, qWdm).


We provide the inference time comparison on the following table, where the inference time of our method to generate an 8-frame egocentric video is 9.06 second, which is comparable with other baselines. We believe it is feasible to use our model in offline applications to generate egocentric videos from the exocentric videos, such as capturing exocentric cooking videos and generating corresponding egocentric videos offline for cooking skills learning. Improving the inference speed towards real-time is very promising and we leave it as future works. We will include the inference time comparison in our final paper.

*Tab. 2 Inference time of our method in comparison with baselines.*
|                         | Ours | SVD  | SD   | PixelNeRF |
|-------------------------|------|------|------|-----------|
| Inference Time (second) | 9.06 | 4.26 | 6.91 | 5.65      |

---

> ### Comment · Area_Chair_baFY · 2024-08-08
> **Author-Reviewer Discussion Begins!**
>
> Dear authors,
>
> Thanks for your hard work and detailed responses.
>
> Dear reviewers,
>
> the authors have responded to your questions and comments, please read them and further provide feedback: Whether your concerns addressed? After reading all the reviews and responses, are there more questions about the clarity, contributions, results, etc?
>
> Thanks!
>
> Best, Your AC

---

> > ### Comment · Area_Chair_baFY · 2024-08-12
> > **Reminder: author-reviewer discussion will end soon!**
> >
> > Dear reviewers,
> >
> > The authors have responded to your questions and comments. Please propose your post-rebuttal comments.
> >
> > Thank Reviewer ANQv for your positive discussions!
> >
> > Best, Your AC

---

### Decision · Program_Chairs · 2024-09-25

**Decision:**

Accept (poster)

**Comment:**

The paper introduces Exo2Ego-V, a diffusion-based video generation method for daily-life skilled human activities with sparse 4-view exocentric viewpoints. It proposes a multi-view exocentric encoder, designing an exocentric-to-egocentric view translation prior, and introducing temporal attention layers. It performs well and shows improvements. Most concerns are addressed, though there is also some discussion about the difference between this work and ReconFusion. After reading the paper, reviews, and responses, the AC agrees that this paper proposed a sound method and made interesting improvements. However, the revision suggested should be made in the final version.